# EnerBridge-DPO: Energy-Guided Protein Inverse Folding with Markov Bridges and Direct Preference Optimization

## Abstract

Designing protein sequences with optimal energetic stability is a key challenge in protein inverse folding, as current deep learning methods are primarily trained by maximizing sequence recovery rates, often neglecting the energy of the generated sequences. This work aims to overcome this limitation by developing a model that directly generates low-energy, stable protein sequences. We propose EnerBridge-DPO, a novel inverse folding framework focused on generating low-energy, high-stability protein sequences. Our core innovation lies in: First, integrating Markov Bridges with Direct Preference Optimization (DPO), where energy-based preferences are used to fine-tune the Markov Bridge model. The Markov Bridge initiates optimization from an information-rich prior sequence, providing DPO with a pool of structurally plausible sequence candidates. Second, an explicit energy constraint loss is introduced, which enhances the energy-driven nature of DPO based on prior sequences. This enables the model to effectively learn energy representations from a wealth of prior knowledge. It can also directly predict sequence energy values, thereby capturing quantitative features of the energy landscape. Our evaluations demonstrate that EnerBridge-DPO can design protein complex sequences with lower energy while maintaining sequence recovery rates comparable to state-of-the-art models, and accurately predicts $\Delta\Delta G$ values between various sequences.

## 1 Introduction

The inverse protein folding problem seeks to identify amino acid sequences that will reliably fold into a given three-dimensional protein backbone. Recent advances in deep learning, particularly with large-scale structure predictors like AlphaFold 2 Jumper et al. (2021), have created unprecedented opportunities for protein inverse folding. Early methods such as ProteinMPNN Dauparas et al. (2022) and PiFold Gao et al. (2022a) treated this as a one-to-one mapping from structure to sequence, which neglected the inherent diversity of sequences that can form a single backbone. To overcome this limitation, more recent models like LM-Design Zheng et al. (2023), GraDe-IF Yi et al. (2023), and Bridge-IF Zhu et al. (2024) employ advanced generative or iterative strategies to explore this "one-to-many" relationship. After training on extensive datasets, these models now show remarkable performance in both sequence recovery and de novo design.

However, a critical challenge persists: designed sequences must not only be compatible with the target structure but also possess desirable physicochemical properties, particularly low free energy, which is correlated with stability Becktel & Schellman (1987) and function Freire (2001); Norn et al. (2021). Existing mainstream inverse folding models still face several key limitations in generating low-energy sequences: First, current diffusion-based generative models aim to learn a single, often intractable, data distribution Zhang et al. (2024); Lemercier et al. (2024). This can lead to a gap between the generated sequences and the true sequence distribution Igashov et al. (2023); Zhu et al. (2024); Lee et al. (2024), making it difficult to efficiently explore the vast sequence space to find candidates that are both structurally consistent and energetically favorable. Second, many advanced inverse folding models are primarily trained by maximizing metrics such as sequence recovery rate or structural similarity. While these models can produce sequences compatible with the target backbone, they generally do not incorporate protein energy stability as a direct optimization objective.

Consequently, the generated sequences may not be energetically optimal and could even be unstable. To our knowledge, there is currently a lack of a dedicated inverse folding framework capable of directly and end-to-end optimizing both sequence recovery and low-energy fitness simultaneously, thereby generating sequences that are intrinsically stable.

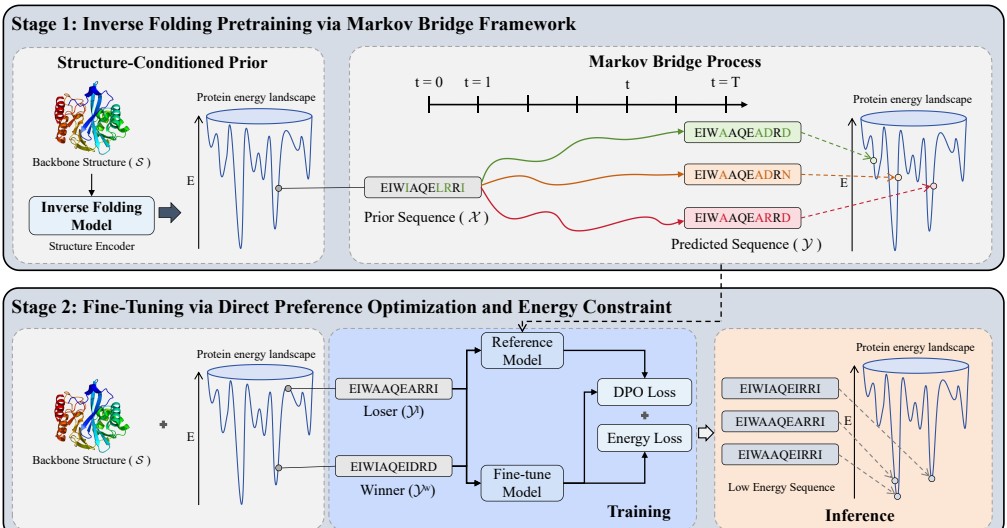

Figure 1: Overview of EnerBridge-DPO. (a) In Stage 1, we pre-train the model on the structure-to-sequence recovery task using a Markov Bridge framework to enhance its structure-sequence alignment capability and sequence diversity. (b) In Stage 2, we fine-tune the model with Bridge-DPO and energy constraints to guide it toward optimizing within a more energetically stable sequence space.

To address these gaps, we introduce EnerBridge-DPO (Energy-Bridged Direct Preference Optimization), a novel inverse folding framework specifically designed for generating low-energy, high-stability protein sequences. As illustrated in Figure 2, our method uniquely integrates two key techniques: First, we establish the foundational architecture of the inverse folding model based on a Markov bridge Zhu et al. (2024); Igashov et al. (2023) and use DPO Rafailov et al. (2023) to fine-tune the pre-trained Markov bridge model. Compared to diffusion models, the Markov bridge begins optimization from an information-rich prior sequence related to the target structure, providing DPO with a candidate pool that already possesses structural plausibility. We construct energy-based preference pairs and adapt the DPO training objective to effectively combine with the Markov bridge's generation process. Second, we introduce an explicit energy constraint loss, which directly requires the model to predict the energy values of sequences, enhancing the energy-driven nature of DPO based on prior sequences. This prompts the model not only to learn the relative energy advantages of sequences but also to understand and fit the quantitative features of the energy landscape. This dual optimization strategy ensures that while EnerBridge-DPO generates low-energy sequences, its internal representations also more closely align with true biophysical principles. Empirical studies show that EnerBridge-DPO outperforms existing baselines on multiple standard benchmarks and excels in designing sequences with low energies.

To summarise, the main contributions of this work are as follows:

- We introduce EnerBridge-DPO, the first inverse folding model that directly utilizes energy constraints within a generative framework to design low-energy protein sequences conforming to specified structural conditions.

- We utilize energy as a preference to innovatively adapt DPO for effective fine-tuning of the Markov Bridge process. Furthermore, an explicit energy constraint loss is introduced, compelling the model to learn and predict quantitative energy features.

- Experimental results demonstrate that EnerBridge-DPO designs protein complex sequences with significantly lower energy and higher stability compared to existing methods, while maintaining comparable sequence recovery and structural validity. The model also accurately predicts $\Delta\Delta G$, highlighting its refined understanding of biophysical principles.

## 2 RELATED WORK

### 2.1 INVERSE PROTEIN FOLDING

Inverse protein folding aims to identify amino acid sequences that fold into a given three-dimensional protein structure. Early deep learning approaches often employed graph neural networks (GNNs) Ingraham et al. (2019); Gao et al. (2022b); Dauparas et al. (2022); Gao et al. (2022a); Tan et al. (2022); Chou et al. (2024), transformers Ingraham et al. (2019); Hsu et al. (2022); Wu et al. (2021), or autoregressive models Ingraham et al. (2019); Hsu et al. (2022) to learn the mapping from structure to sequence. To address error accumulation in autoregressive generation, iterative refinement strategies have emerged. Some methods leverage the knowledge encoded in pre-trained Protein Language Models (PLMs) to refine initially generated sequences, such as LM-Design Zheng et al. (2023) and KW-Design Gao et al. (2023). Recently, discrete diffusion models Austin et al. (2021) have been adapted for sequence generation. GraDe-IF Yi et al. (2023) pioneered the use of denoising diffusion for inverse folding, conditioning the denoising process on structural information. While diffusion models offer a principled way for iterative refinement and capturing diversity, standard formulations often start from a non-informative prior (e.g., uniform noise), potentially limiting efficiency and the ability to leverage strong structural information directly. Markov Bridge models, such as Bridge-IF Zhu et al. (2024), offer an alternative generative framework. They learn a stochastic process between two distributions, allowing the use of an informative, structure-derived prior sequence as the starting point and progressively refining it towards the target native sequence distribution, potentially offering advantages in sample quality and inference efficiency.

### 2.2 INCORPORATING PHYSICAL CONSTRAINTS AND ENERGY FUNCTIONS

A key challenge in protein design is ensuring the physical and chemical viability of generated sequences. Many approaches incorporate physics-based information Norn et al. (2021); Omar et al. (2023); Malbranke et al. (2023), often as a post-processing step. For example, generated sequences might be filtered or rescored using energy functions like Rosetta Rohl et al. (2004) or evaluated using molecular dynamics simulations. While helpful, these two-stage approaches mean the generative model itself isn't directly optimizing for physical properties like energy stability during generation, potentially limiting the effectiveness of the design process. Few methods Zhou et al. (2024); Ren et al. (2025) have successfully integrated energy functions directly into the end-to-end training and optimization loop of deep generative models for protein design. Our work, EnerBridge-DPO, aims to bridge this gap by directly optimizing the generative bridge model for lower energy using preference optimization.

### 2.3 PREFERENCE OPTIMIZATION IN INVERSE FOLDING

Aligning generative models with specific preferences or criteria has been highly successful, particularly in Large Language Models (LLMs). In contrast to its extensive exploration in NLP, the application of DPO to protein inverse folding design has been less explored. Park et al. Park et al. (2024) proposed a diversity-regularized DPO method to address issues of sequence repetition and folding failures in peptide inverse folding. Their approach fine-tunes the ProteinMPNN model by introducing a regularization term to DPO, effectively enhancing the diversity and stability of the generated sequences. Similarly, Xue et al. Xue et al. (2025) proposed a new strategy to optimize protein sequence design via DPO, aiming to improve the "designability" of a sequence by using AlphaFold's pLDDT score as the guiding signal for DPO training. However, these applications of DPO have primarily focused on enhancing the capacity of designed sequences to fold into a target structure, whereas our work explores lowering the energy of the designed sequences.

## 3 METHODOLOGY

### 3.1 PRELIMINARY

The Inverse Folding problem seeks to generate an amino-acid sequence $\mathcal{Y} = (y_1, y_2, ..., y_L)$ that will reliably fold into a given protein backbone structure $\mathcal{S} = (s_1, s_2, ..., s_L)$, where each residue's coordinates $s_i \in \mathbb{R}^{4 \times 3}$ typically include the N, C-$\alpha$, and C atoms (with O atoms optional), and $L$ denotes

the length of the backbone. In an ideal design, the proposed sequence should not only reproduce the target backbone under structure prediction but also possess low physical energy and favorable biochemical properties. To formalize this, we treat our generative model as a prior $p_\theta(\mathcal{Y} \mid \mathcal{S})$, and introduce an energy function $\mathcal{E}(\mathcal{S}, \mathcal{Y})$ which serves as a likelihood score or potential energy reflecting the physical plausibility of the sequence–structure pair. Under this formulation, the posterior distribution over sequences given the backbone is

$$p(\mathcal{Y} \mid \mathcal{S}) \propto p_\theta(\mathcal{Y} \mid \mathcal{S}) \exp\left(-\alpha\mathcal{E}(\mathcal{S}, \mathcal{Y})\right) \tag{1}$$

where $\alpha > 0$ governs the relative weight of the energy term. The optimal sequence $\mathcal{Y}^*$—balancing the learned prior and physical constraints—is obtained by maximizing the following objective:

$$\mathcal{Y}^* = \arg\max_{\mathcal{Y}} \left[\log p_\theta(\mathcal{Y} \mid \mathcal{S}) - \alpha\mathcal{E}(\mathcal{S}, \mathcal{Y})\right] \tag{2}$$

This energy-aware framework enables the design of sequences that are both structurally faithful and thermodynamically stable.

### 3.2 Pre-trained Markov Bridge Model for Inverse Folding

The foundation of EnerBridge-DPO is a generative model designed to capture the complex relationship between protein backbone structures and their corresponding amino acid sequences. This model leverages the Markov Bridge framework to learn the probabilistic transition from an initial sequence proposal, derived directly from the input structure, to the target native sequence distribution.

#### 3.2.1 Structure-Conditioned Prior

Unlike traditional diffusion models that start from random noise, our approach begins with an informative prior sequence $\mathcal{X}$. This prior is generated by PiFold, as an expressive structure encoder $\mathcal{E}$, which maps the input backbone structure $\mathcal{S}$ to a discrete sequence $\mathcal{X} = \mathcal{E}(\mathcal{S})$. This encoder is pre-trained on large-scale structure-sequence datasets to predict plausible sequences directly from structures. This deterministic mapping provides a strong sequence prior for the subsequent refinement process.

#### 3.2.2 Markov Bridge Process

We establish a discrete-time Markov Bridge process $(\boldsymbol{z}_t)_{t=0}^{T}$ connecting the distribution of the prior sequence $p_x(\mathcal{X})$ and the distribution of the native sequence $p_y(\mathcal{Y})$. The process starts at $\boldsymbol{z}_0 = \mathcal{X}$ and is designed to end at $\boldsymbol{z}_T = \mathcal{Y}$ that satisfies

$$p(\boldsymbol{z}_t|\boldsymbol{z}_0, \boldsymbol{z}_1, \ldots, \boldsymbol{z}_{t-1}, \boldsymbol{\mathcal{Y}}) = p(\boldsymbol{z}_t|\boldsymbol{z}_{t-1}, \boldsymbol{\mathcal{Y}}). \tag{3}$$

To pin the process at the end point $\boldsymbol{z}_T = \mathcal{Y}$, we have an additional requirement

$$p(\boldsymbol{z}_T = \mathcal{Y}|\boldsymbol{z}_{T-1}, \mathcal{Y}) = 1. \tag{4}$$

We assume that $p(\cdot)$ is categorical distributions with a finite sample space $\{1, ..., K\}$ and represent data points as one-hot vectors: $\mathcal{X}, \mathcal{Y}, \boldsymbol{z}_t \in \{0, 1\}$. The forward process gradually transforms $\mathcal{X}$ towards $\mathcal{Y}$ using a transition matrix $\boldsymbol{Q}_t$:

$$\boldsymbol{Q}_t := \boldsymbol{Q}_t(\mathcal{Y}) = \gamma_t \boldsymbol{I}_K + (1 - \gamma_t)\mathcal{Y}\boldsymbol{1}_K^\top \tag{5}$$

where $\gamma_t$ is a noise schedule transitioning from $\gamma_0 = 1$ to $\gamma_{T-1} = 0$. The core of the generative model lies in learning the reverse process, which approximates the target sequence $\mathcal{Y}$ at each step $t$ given the intermediate state $\boldsymbol{z}_t$ and the structure $\mathcal{S}$.

#### 3.2.3 Model Architecture and Training

We utilize a pre-trained PLM, such as ESM Rives et al. (2021), as the backbone for approximating the reverse bridge process. To effectively condition the PLM on both the time step $t$ and the structural information $\mathcal{S}$, we adapt its architecture using AdaLN-Bias Peebles & Xie (2023) and structural adapters (cross-attention). These modifications allow the PLM to leverage its learned evolutionary knowledge while integrating the specific temporal and structural context of the bridge process, all while maintaining parameter efficiency by keeping the base PLM weights frozen. The pre-training minimizes a

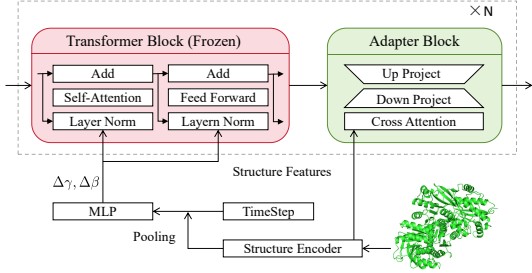

Figure 2: Architecture of Markov Bridge Model.

loss function derived from the Markov Bridge framework, aiming to accurately predict the target sequence $\mathcal{Y}$. We utilize the simplified reparameterized objective function derived in Bridge-IF for more effective training:

$$\mathcal{L}_t(\theta) = \lambda_t \mathbb{E}_{p(\boldsymbol{z}_t|\mathcal{X},\mathcal{Y})}[-v_t \mathcal{Y}^T \log \varphi_\theta(\boldsymbol{z}_t, \mathcal{S}, t)] \tag{6}$$

where $\varphi_\theta(\boldsymbol{z}_t, \mathcal{S}, t)$ is the PLM's prediction of the target sequence, $v_t$ indicates if the token needs refinement, and $\lambda_t$ is a weighting factor. During inference, the model starts with the prior sequence $\boldsymbol{z}_0 = \mathcal{E}(\mathcal{S})$ and iteratively refines it using the learned reverse process conditioned on $\mathcal{S}$ and $t$, ultimately generating the final designed sequence $\boldsymbol{z}_T$.

### 3.3 Bridge-DPO

We employ DPO to fine-tune the model, explicitly biasing it towards generating sequences with lower predicted energy states, particularly for protein complexes. This stage aims to integrate physical realism, specifically energy stability, directly into the generative process.

#### 3.3.1 Preference Data Generation

We leverage the measured energy values as the preference oracle. For a given protein backbone structure $\mathcal{S}$, we generate or select pairs of candidate sequences $(\mathcal{Y}^w, \mathcal{Y}^l)$. A pair is included in our preference dataset $\mathcal{D}_{energy} = \{(\mathcal{S}, \mathcal{Y}^w, \mathcal{Y}^l)\}$ if the experiment measures a lower energy for sequence $\mathcal{Y}^w$ (winning/preferred sequence) compared to sequence $\mathcal{Y}^l$ (losing/less preferred sequence), i.e., $Energy(\mathcal{Y}^w \mid \mathcal{S}) < Energy(\mathcal{Y}^l \mid \mathcal{S})$.

#### 3.3.2 DPO Objective for Markov Bridges

We adapt the DPO objective to the context of our Markov Bridge model. The core idea is to maximize the likelihood of preferred sequences $(\mathcal{Y}^w)$ while minimizing the likelihood of less preferred sequences $(\mathcal{Y}^l)$, relative to a reference model $\varphi_{ref}$. The reference model $\varphi_{ref}$ is the pre-trained Markov Bridge model obtained from the initial training phase. The DPO loss function for fine-tuning our Bridge model $\varphi_\theta$ is derived analogously to the objectives in DPO Rafailov et al. (2023) and Diffusion-DPO Wallace et al. (2024):

$$\mathcal{L}_{\text{Bridge-DPO}}(\theta) = -\mathbb{E}_{(\mathcal{Y}^w,\mathcal{Y}^l)\sim\mathcal{D},t\sim\mathcal{U}(0,T),\boldsymbol{z}_t^w\sim q(\boldsymbol{z}_t^w|\mathcal{X},\mathcal{S}),\boldsymbol{z}_t^l\sim q(\boldsymbol{z}_t^l|\mathcal{X},\mathcal{S})}$$
$$\log \sigma(-\beta T\omega(\lambda_t)(\|\mathcal{Y}^w - \varphi_\theta(\boldsymbol{z}_t^w,t)\|_2^2 - \|\mathcal{Y}^w - \varphi_{\text{ref}}(\boldsymbol{z}_t^w,t)\|_2^2) \tag{7}$$
$$- (\|\mathcal{Y}^l - \varphi_\theta(\boldsymbol{z}_t^l,t)\|_2^2 - \|\mathcal{Y}^l - \varphi_{\text{ref}}(\boldsymbol{z}_t^l,t)\|_2^2))$$

Here, $\varphi_\theta(\mathcal{Y}|\mathcal{S})$ represents the probability (or a related measure like likelihood derived from the bridge process) assigned by the model $\varphi_\theta$ to sequence $\mathcal{Y}$ given structure $\mathcal{S}$. $\beta$ is a hyperparameter that controls the strength of the deviation from the reference model $\varphi_{ref}$. A higher $\beta$ imposes a stronger penalty for diverging from the initial pre-trained model, ensuring that the learned preference alignment does not drastically compromise the model's original capabilities. A detailed derivation can be seen in Appendix C.

In the context of Markov Bridge models, the likelihood ratio term can be related to the difference in the model's internal predictions or losses during the bridge process, similar to how Diffusion-DPO relates it to denoising errors. For instance, using the simplified cross-entropy loss formulation,

the DPO objective implicitly encourages the model $\varphi_\theta$ to have lower prediction errors (or higher probabilities) for the preferred sequence $\mathcal{Y}^w$ compared to $\mathcal{Y}^l$, relative to the reference model $\varphi_{ref}$, across the bridge timesteps.

### 3.4 ENERGY-CONSTRAINED LOSS

To explicitly enforce low-energy designs, we integrate a BA-DDG-based Jiao et al. (2024) energy constraint into our fine-tuning stage. First, for each winner-loser pair of a given protein, assuming their backbone structures are identical, we compute the predicted binding free-energy change $\widehat{\Delta\Delta G}$ directly from inverse-folding log-likelihoods via Boltzmann Alignment. Specifically, we first calculate the binding free energy for the winner and loser sequences individually, and then compute the difference between these two values. For each protein (whether it is the winner or loser), its binding free energy arises from the energy difference between the bound and unbound states. In our model, this energy difference is represented by the sequence likelihood:

$$\widehat{\Delta\Delta G} = -k_{\mathrm{B}}T \cdot \Big( \log \frac{p(\mathcal{Y}^{\mathrm{winner}}_{\mathrm{bnd}} \mid \mathcal{S}_{\mathrm{bnd}})}{p(\mathcal{Y}^{\mathrm{winner}}_{\mathrm{unbnd}} \mid \mathcal{S}_{\mathrm{unbnd}})} - \log \frac{p(\mathcal{Y}^{\mathrm{loser}}_{\mathrm{bnd}} \mid \mathcal{S}_{\mathrm{bnd}})}{p(\mathcal{Y}^{\mathrm{loser}}_{\mathrm{unbnd}} \mid \mathcal{S}_{\mathrm{unbnd}})} \Big) \tag{8}$$

where $k_{\mathrm{B}}T$ is treated as a learnable scaling factor and "bnd" means bound. We then define the energy constraint loss as the mean absolute error between predicted and true $\Delta\Delta G$ over the labeled set $\mathcal{D}$:

$$\mathcal{L}_{\mathrm{energy}}(\theta) = \frac{1}{|\mathcal{D}|} \sum_{(s, y_{\mathrm{winner}}, y_{\mathrm{loser}}, \Delta\Delta G) \in \mathcal{D}} |\widehat{\Delta\Delta G} - \Delta\Delta G| \tag{9}$$

### 3.5 OVERALL TRAINING OBJECTIVE

During the fine-tuning phase of EnerBridge-DPO, the overall training loss $\mathcal{L}_{\mathrm{total}}(\theta)$ is formulated as a weighted sum of the Bridge-DPO loss ($\mathcal{L}_{\mathrm{Bridge-DPO}}(\theta)$) and the energy-constrained loss ($\mathcal{L}_{\mathrm{energy}}(\theta)$):

$$\mathcal{L}_{\mathrm{total}}(\theta) = \mathcal{L}_{\mathrm{Bridge-DPO}}(\theta) + 0.5 * \mathcal{L}_{\mathrm{energy}}(\theta) \tag{10}$$

## 4 EXPERIMENTS

### 4.1 EXPERIMENTAL PROTOCOL

#### 4.1.1 DATASETS

We conducted benchmarking on the following datasets:

**MPNN** Dauparas et al. (2022): All sequences are clustered at $30\%$ sequence identity, resulting in 25,361 distinct clusters. Following ProteinMPNN's setup, we split these clusters into three disjoint sets: a training set (23,358 clusters), a validation set (1,464 clusters), and a test set, ensuring that neither the target chains nor any chains from their biological assemblies appear across multiple splits.

**BindingGym** Lu et al. (2024): BindingGym contains 10M mutational data points. For each protein in BindingGYM, we select the top $10\%$ and bottom $10\%$ of mutants based on their Deep Mutational Scanning (DMS) scores and randomly pair them to construct preference pairs.

**SKEMPI** Jankauskaitė et al. (2019): Following the methods of Luo et al. (2023) and Wu & Li (2024), we divided the dataset into 3 parts based on structure, ensuring that each part contains unique protein complexes. Based on this division, a three-fold cross-validation process was performed. For each protein complex, we selected mutations from the top $30\%$ and bottom $30\%$ ranked by binding energy and randomly paired them to construct preference pairs.

The construction of the preference dataset is detailed in Appendix E.

### 4.1.2 Implementation Details

We pre-trained the model on the MPNN Dataset. Subsequent fine-tuning was performed using preference pairs constructed from the BindingGym Dataset and SKEMPI Dataset. We use the cosine schedule Nichol & Dhariwal (2021) with number of timestep T = 25 for the noise scheduling. For pre-training, the model was trained for 50 epochs on an NVIDIA 4090 GPU with a batch size of 40,000 residues, an initial learning rate of 0.001, and Adam optimizer Kingma & Ba (2014) with noam learning rate scheduler Vaswani et al. (2017) was used. During fine-tuning, we maintained the same architecture but reduced the learning rate to $1 \times e^{-5}$, while extending training to 150 epochs for enhanced parameter refinement. All experiments are conducted on a computing cluster with CPUs of Intel(R) Xeon(R) Gold 6144 CPU of 3.50GHz and two NVIDIA GeForce RTX 4090 24GB GPU.

Table 1: Results comparison on the MPNN dataset. The **best** and suboptimal results are labeled with bold and underlined.

| Model | Perplexity ↓ | | | | Recovery % ↑ | | | |
|---|---|---|---|---|---|---|---|---|
| | $L < 100$ | $100 \leq L < 500$ | $500 \leq L < 1000$ | Full | $L < 100$ | $100 \leq L < 500$ | $500 \leq L < 1000$ | Full |
| GraphTrans | 6.67 | 5.59 | 5.58 | 5.72 | 41.39 | 45.71 | 45.36 | 45.40 |
| GCA | 6.41 | 5.33 | 5.30 | 5.45 | 43.45 | 46.93 | 46.52 | 46.58 |
| SructGNN | 6.50 | 5.29 | 5.21 | 5.43 | 42.51 | 47.72 | 47.54 | 47.36 |
| AlphaDesign | 6.77 | 5.33 | 5.21 | 5.49 | 43.62 | 48.04 | 47.81 | 47.81 |
| GVP | 6.24 | 4.74 | 4.54 | 4.90 | 45.68 | 51.10 | 51.46 | 50.75 |
| PiFold | 6.05 | 4.18 | 3.93 | 4.38 | 48.54 | 55.62 | 56.47 | 55.17 |
| ProteinMPNN | 5.63 | 4.09 | 3.83 | 4.25 | 50.04 | 57.09 | 59.04 | 57.28 |
| LMDesign | 4.60 | 4.06 | 3.99 | 4.06 | 55.07 | 57.18 | 57.99 | 58.14 |
| Bridge-IF | **4.44** | 3.91 | 3.74 | 3.89 | **58.42** | 60.15 | 60.74 | 60.56 |
| EnerBridge-DPO | 4.51 | **3.88** | **3.68** | **3.87** | 57.90 | **60.41** | **61.14** | **60.91** |

### 4.1.3 Baselines

We evaluate the performance of our model on two tasks: protein inverse folding and the change in binding free energy ($\Delta\Delta G$) prediction. For the protein inverse folding task, we compare EnerBridge-DPO against several state-of-the-art baselines, including GraphTrans Wu et al. (2021), StructGNN Chou et al. (2024), GVP Jing et al. (2020), GCA Tan et al. (2023), AlphaDesign Gao et al. (2022b), ProteinMPNN Dauparas et al. (2022), PiFold Gao et al. (2022a), LM-Design Zheng et al. (2023), GraDe-IF Yi et al. (2023), and Bridge-IF Zhu et al. (2024). For the $\Delta\Delta G$ prediction task, we compare our approach with state-of-the-art supervised methods, including DDGPred Shan et al. (2022), MIF-Network Yan et al. (2020), RDE-Network Luo et al. (2023), DiffAffinity Lin et al. (2022), Prompt-DDG Wu et al. (2024), ProMIM Mo et al. (2024), Surface-VQMAE Wu & Li (2024), and BA-DDG Jiao et al. (2024).

### 4.1.4 Evaluation

We use perplexity and recovery rate to evaluate the generation quality of the inverse folding task. Following previous studies Ingraham et al. (2019), we report perplexity and median recovery rate under four settings: short proteins (length < 100), medium proteins (100≤length <500), long proteins (500≤length <1000), and full proteins. Furthermore, we also report the energy of the resulting sequences. In addition to the mean and standard deviation, we also use $\text{ZScore} = e^{\text{mean}_1(x)(\frac{x - \text{mean}_2(x)}{\text{std}_2(x)})}$ to score the models, where $\text{mean}_1$ is the mean value for a specific model across different methods, and $\text{mean}_2$ is the mean value for a specific method across different models. To comprehensively evaluate the performance of $\Delta\Delta G$ prediction, we use a total of seven overall metrics, including 5 overall metrics: (1) Pearson correlation coefficient, (2) Spearman rank correlation coefficient, (3) minimized RMSE, (4) minimized MAE, (5) AUROC, and 2 per-structure metrics: (6) Per-structure Pearson correlation coefficient and (7) Per-structure Spearman correlation coefficients.

### 4.2 Inverse Folding

As shown in Table 1, experimental results indicate that EnerBridge-DPO achieves performance comparable to Bridge-IF in terms of sequence recovery rate and perplexity. This suggests that the energy preference fine-tuning via DPO does not negatively impact its fundamental inverse folding fidelity.

Despite performing comparably to Bridge-IF on general inverse folding metrics, the core advantage of EnerBridge-DPO lies in the energetic optimization of sequences designed for protein complexes. To validate this, we specially selected 26 protein complex structures with distinct chains from the test set to serve as test cases. For these specific complexes, we employed EnerBridge-DPO and other baseline models to design sequences. Then, using three unsupervised energy prediction models (including FoldX Delgado et al. (2019), Rosetta Alford et al. (2017) and BA-Cycle Jiao et al. (2024)), we assessed the predicted binding free energy or stability of these designed sequences.

The results, as shown in Table 2, clearly demonstrate that sequences generated by EnerBridge-DPO for these selected protein complexes have significantly lower predicted energy values than all comparative models. This strongly validates the effectiveness of the energy preference learning introduced during the DPO fine-tuning phase, enabling EnerBridge-DPO to specifically optimize and generate more energetically stable and physicochemically sound protein complex sequences.

Table 2: Comparison of energy for sequences designed by different models. The **best** and suboptimal results are labeled with bold and underlined.

| Model | FoldX | | Rosetta | | BA-Cycle | | ZScore ↓ |
|---|---|---|---|---|---|---|---|
| | Mean ↓ | Std ↓ | Mean ↓ | Std ↓ | Mean ↓ | Std ↓ | |
| GraphTrans | 243.78 | 128.25 | 2813.42 | 3193.46 | 129.19 | 148.83 | 1.82 |
| StructGNN | 235.33 | 133.79 | 2849.01 | 3212.60 | 121.49 | 153.97 | 1.65 |
| GCA | 233.97 | 146.12 | 2842.93 | 3206.22 | 126.77 | 148.22 | 1.74 |
| GVP | 223.32 | 125.85 | 2884.59 | 3283.36 | 125.90 | 141.56 | 1.76 |
| AlphaDesign | 217.01 | 126.02 | 2880.81 | 3273.53 | 128.59 | 151.01 | 1.72 |
| PiFold | 176.69 | 117.97 | 2814.60 | 3274.96 | 120.65 | 122.20 | 0.86 |
| ProteinMPNN | 163.58 | 109.53 | **2697.03** | 3362.60 | 112.32 | 125.50 | 0.47 |
| LMDesign | 202.20 | 295.54 | 3160.13 | 3447.61 | 79.43 | **92.83** | 1.52 |
| Bridge-IF | 163.02 | 98.63 | 2807.27 | **3116.33** | 83.65 | 112.37 | 0.41 |
| EnerBridge-DPO | **130.03** | **76.31** | 2780.20 | 3157.82 | **77.46** | 96.79 | **0.25** |

Table 3: Comparison of 3-fold cross-validation on the SKEMPI v2 dataset. The **best** and suboptimal results are labeled with bold and underlined.

| Method | Per-Structure | | Overall | | | | |
|---|---|---|---|---|---|---|---|
| | Pearson ↑ | Spear. ↑ | Pearson ↑ | Spear. ↑ | RMSE ↓ | MAE ↓ | AUROC ↑ |
| DDGPred | 0.3750 | 0.3407 | 0.6580 | 0.4687 | 1.4998 | 1.0821 | 0.6992 |
| MIF-Network | 0.3965 | 0.3509 | 0.6523 | 0.5134 | 1.5932 | 1.1469 | 0.7329 |
| RDE-Network | 0.4448 | 0.4010 | 0.6447 | 0.5584 | 1.5799 | 1.1123 | 0.7454 |
| DiffAffinity | 0.4220 | 0.3970 | 0.6609 | 0.5560 | 1.5350 | 1.0930 | 0.7440 |
| Prompt-DDG | 0.4768 | 0.4321 | 0.6764 | 0.5936 | 1.5308 | 1.0839 | 0.7567 |
| ProMIM | 0.4640 | 0.4310 | 0.6720 | 0.5730 | 1.5160 | 1.0890 | 0.7600 |
| Surface-VQMAE | 0.4694 | 0.4324 | 0.6482 | 0.5611 | 1.5876 | 1.1271 | 0.7469 |
| BA-DDG | **0.5603** | **0.5195** | 0.7319 | 0.6433 | 1.4426 | **1.0044** | 0.7769 |
| EnerBridge-DPO | 0.4981 | 0.4666 | **0.7487** | **0.6447** | **1.4257** | 1.0185 | **0.7780** |

## 4.3 PROTEIN ENERGY PREDICTION

As shown in Table 3, our results indicate that EnerBridge-DPO achieves $\Delta\Delta G$ prediction performance comparable to that of the BA-DDG predictor. Both methods demonstrate strong correlations with experimental data and low error rates, positioning them at the state-of-the-art for this task. This suggests that EnerBridge-DPO has effectively internalized the ability to predict energy changes, rather than merely learning to generate sequences that are likely to be low in energy according to energy-constrained loss. The model's capacity to accurately predict $\Delta\Delta G$ values underscores its refined understanding of the underlying biophysical principles governing protein interactions and stability. Notably, in the per-structure performance evaluation, while EnerBridge-DPO's performance remains robust, it is slightly outperformed by BA-DDG, a method specifically designed for this task. We hypothesize that this difference may be related to the data sampling strategy employed

during our DPO training phase. To construct preference pairs, we sampled from the original data distribution, which might have led to disparities in the amount of data per fold during the three-fold cross-validation training.

Table 4: Ablation study of key design choices on the MPNN dataset. The **best** and suboptimal results are labeled with bold and underlined.

| w/o DPO | w/o Energy | Perplexity ↓ | | | | Recovery % ↑ | | | |
|---|---|---|---|---|---|---|---|---|---|
| | | $L < 100$ | $100 \leq L < 500$ | $500 \leq L < 1000$ | Full | $L < 100$ | $100 \leq L < 500$ | $500 \leq L < 1000$ | Full |
| | ✓ | 4.59 | 3.99 | 3.97 | 4.06 | 56.05 | 60.10 | 61.10 | 60.18 |
| ✓ | | 6.10 | 4.04 | 4.08 | 4.16 | 54.81 | 59.50 | 60.12 | 59.52 |
| | | **4.51** | **3.88** | **3.68** | **3.87** | **57.90** | **60.41** | **61.14** | **60.91** |

Table 5: Ablation study of key design choices on the SKEMPI v2 dataset. The **best** and suboptimal results are labeled with bold and underlined.

| w/o DPO | w/o Energy | Per-Structure | | Overall | | | | |
|---|---|---|---|---|---|---|---|---|
| | | Pearson ↑ | Spear. ↑ | Pearson ↑ | Spear. ↑ | RMSE ↓ | MAE ↓ | AUROC ↑ |
| | ✓ | 0.3493 | 0.2975 | 0.4134 | 0.4259 | 1.9005 | 1.3475 | 0.6784 |
| ✓ | | 0.4893 | 0.4623 | 0.7400 | 0.6227 | 1.4539 | 1.0360 | 0.7720 |
| | | **0.4981** | **0.4666** | **0.7487** | **0.6447** | **1.4257** | **1.0185** | **0.7780** |

## 4.4 ABLATION STUDIES

To validate the contributions of the key components within the EnerBridge-DPO framework, we conducted a series of ablation studies. We designed the following model variants for comparison: (1) w/o DPO: This model is fine-tuned using only the energy-constrained loss $\mathcal{L}_{\text{energy}}$ without the energy preference-based DPO process. (2) w/o Energy: This model is fine-tuned using only the energy preference-based DPO process (via $\mathcal{L}_{\text{Bridge}-\text{DPO}}$) without the explicit energy-constrained loss.

**w/o DPO**: From Table 4, we observe that without the DPO process, the core inverse folding metrics are significantly worse. This is because the primary role of DPO is to adjust the generative model's output distribution to align with preferences. With only an energy regression loss, the model might impair the pre-trained model's strong sequence generation capabilities by attempting to forcibly fit energy values.

**w/o Energy**: From Table 5, We observe that without the explicit energy-constrained loss, the model performs very poorly on all $\Delta\Delta G$ prediction metrics. It illustrates that the model might learn to generate lower-energy sequences but will be unable to accurately quantify these energy differences or predict specific $\Delta\Delta G$ values.

## 5 CONCLUSION

In this paper, we introduce EnerBridge-DPO, a novel framework that addresses the critical challenge of generating energetically stable protein sequences in inverse folding. EnerBridge-DPO uniquely combines Markov bridges with Direct Preference Optimization based on sequence energies, and employs an explicit energy constraint to prompt the model to capture the energy values of generated sequences. Experimental results demonstrate that EnerBridge-DPO successfully designs protein sequences, particularly for complexes, with significantly reduced predicted energies and enhanced stability. Notably, this improved energy performance is achieved while maintaining sequence recovery rates comparable to state-of-the-art methods and accurately predicting $\Delta\Delta G$ values. Future work will explore ensuring the diversity and quality of DPO preference pairs and extending the validation of model performance to a broader range of protein families and more complex structural scenarios.

# 6 ETHICS STATEMENT

This work adheres to the ICLR Code of Ethics. In this study, no human subjects or animal experimentation was involved. All datasets used, including MPNN, BindingGym, and SKEMPI were sourced in compliance with relevant usage guidelines, ensuring no violation of privacy. We have taken care to avoid any biases or discriminatory outcomes in our research process. No personally identifiable information was used, and no experiments were conducted that could raise privacy or security concerns. We are committed to maintaining transparency and integrity throughout the research process.

# 7 REPRODUCIBILITY STATEMENT

We have made every effort to ensure that the results presented in this paper are reproducible. All code has been submitted in the Supplementary Material to facilitate replication and verification. The experimental setup, including training steps, model configurations, and hardware details, is described in detail in the paper. We believe these measures will enable other researchers to reproduce our work and further advance the field.

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

## A  THE USE OF LARGE LANGUAGE MODELS

During the preparation of this manuscript, we utilized Gemini-2.5-pro as an assistive tool for language enhancement. The primary use of these models was to improve the grammar, clarity, and readability of the text. All scientific ideas, experimental results, and conclusions were conceived and written by the human authors, who retain full responsibility for the final content of this paper.

## B  ALGORITHMS

---

**Algorithm 1** DPO Fine-tuning for EnerBridge-DPO

---

**Input**: Pre-trained reference model $\varphi_{ref}$, initialize policy model $\varphi_\theta$, preference dataset $\mathcal{D}_{energy} = \{(\mathcal{S}, \mathcal{Y}^w, \mathcal{Y}^l)\}$

**Output**: Fine-tuned model $\varphi_\theta$

1: **repeat**
2:     Sample $(\mathcal{S}, \mathcal{Y}^w, \mathcal{Y}^l)$ from $\mathcal{D}_{energy}$
3:     Sample timestep $t \sim \mathcal{U}(0, T)$
4:     Generate prior sequence $\mathcal{X} \leftarrow \text{StructureEncoder}(\mathcal{S})$
5:     Generate $z_t^w \sim q(z_t | \mathcal{X}, \mathcal{S}, \mathcal{Y}^w)$
6:     Generate $z_t^l \sim q(z_t | \mathcal{X}, \mathcal{S}, \mathcal{Y}^l)$
7:     $\text{err}_\theta^w \leftarrow ||\mathcal{Y}^w - \varphi_\theta(z_t^w, \mathcal{S}, t)||_2^2$
8:     $\text{err}_{ref}^w \leftarrow ||\mathcal{Y}^w - \varphi_{ref}(z_t^w, \mathcal{S}, t)||_2^2$
9:     $\text{err}_\theta^l \leftarrow ||\mathcal{Y}^l - \varphi_\theta(z_t^l, \mathcal{S}, t)||_2^2$
10:    $\text{err}_{ref}^l \leftarrow ||\mathcal{Y}^l - \varphi_{ref}(z_t^l, \mathcal{S}, t)||_2^2$
11:    $\text{diff\_winner} \leftarrow \text{err}_\theta^w - \text{err}_{ref}^w$
12:    $\text{diff\_loser} \leftarrow \text{err}_\theta^l - \text{err}_{ref}^l$
13:    $\text{dpo\_term} \leftarrow \text{diff\_winner} - \text{diff\_loser}$
14:    $L_{DPO} \leftarrow -\log \sigma(-\beta_{dpo} \cdot \text{dpo\_term})$
15:    Update $\varphi_\theta$ using gradient of $L_{DPO}$
16: **until** convergence
17: **return** $\varphi_\theta$

---

**Algorithm 2** Sampling

---

**Input**: Starting point $\mathcal{S} \sim p_\mathcal{S}$, structure encoder $\mathcal{E}$, neural network $\varphi_\theta$

**Output**: Generated sequence $z_T$

1: $z_0 \leftarrow \mathcal{E}(\mathcal{S})$
2: **for** $t$ in $0, \ldots, T - 1$ **do**
3:     $\hat{y} \leftarrow \varphi_\theta(z_t, t)$
4:     $q_\theta(z_{t+1} | z_t) \leftarrow \text{Cat}(z_{t+1}; Q_t(\hat{y})z_t)$
5:     $z_{t+1} \sim q_\theta(z_{t+1} | z_t)$
6: **end for**
7: **return** $z_T$

---

## C  DPO FOR MARKOV BRIDGE MODELS

The goal is to adapt the Direct Preference Optimization (DPO) framework for Markov Bridge models used in protein inverse folding. We have a fixed dataset $\mathcal{D}_{energy} = \{(\mathcal{S}, \mathcal{Y}^w, \mathcal{Y}^l)\}$ where each example contains a protein backbone structure $\mathcal{S}$, a preferred (winner) sequence $\mathcal{Y}^w$, and a dispreferred (loser) sequence $\mathcal{Y}^l$, generated from a reference Markov Bridge model $\phi_{ref}$. We aim to learn a new model $\phi_\theta$ that is aligned with these energy-based preferences.

### C.1  STARTING POINT: THE RLHF OBJECTIVE AND ITS DPO REFORMULATION

The general objective in Reinforcement Learning from Human Feedback (RLHF), which DPO aims to solve more directly, is to optimize a policy $p_\theta$ to maximize a reward function $r(c, x_0)$ (where $c$ is

context and $x_0$ is generation) while regularizing its deviation from a reference policy $p_{ref}$ using a KL-divergence term:

$$\max_{p_\theta} \mathbb{E}_{c \sim \mathcal{D}_c, x_0 \sim p_\theta(x_0|c)}[r(c, x_0)] - \beta \mathbb{D}_{\text{KL}}[p_\theta(x_0|c)||p_{ref}(x_0|c)] \quad (11)$$

DPO shows that the optimal solution $p_\theta^*(x_0|c)$ can be written as:

$$p_\theta^*(x_0|c) = \frac{1}{Z(c)} p_{ref}(x_0|c) \exp\left(\frac{1}{\beta} r(c, x_0)\right) \quad (12)$$

This allows rewriting the reward function $r(c, x_0)$ in terms of $p_\theta^*$ and $p_{ref}$. Substituting this into the Bradley-Terry model for preferences $p(x_0^w > x_0^l|c) = \sigma(r(c, x_0^w) - r(c, x_0^l))$, leads to the DPO loss:

$$L_{DPO}(\theta) = -\mathbb{E}_{(c, x_0^w, x_0^l)} \left[\log \sigma \left(\beta \log \frac{p_\theta(x_0^w|c)}{p_{ref}(x_0^w|c)} - \beta \log \frac{p_\theta(x_0^l|c)}{p_{ref}(x_0^l|c)}\right)\right] \quad (13)$$

## C.2 ADAPTING TO MARKOV BRIDGE PROCESS

For generative models like Markov Bridge, the likelihood $p_\theta(\mathcal{Y}|\mathcal{S})$ is often intractable as it requires marginalizing over all possible generative paths $z_{0:T}$ (where $z_0 = \mathcal{X}$, the prior sequence from structure $\mathcal{S}$, and $z_T = \mathcal{Y}$, the final generated sequence).

We define the objective over the entire path $z_{0:T}$. The RLHF objective becomes:

$$\max_{p_\theta} \mathbb{E}_{\mathcal{S} \sim \mathcal{D}_\mathcal{S}, z_{0:T} \sim p_\theta(z_{0:T}|\mathcal{S})}[r(\mathcal{S}, \mathcal{Y})] - \beta \mathbb{D}_{\text{KL}}[p_\theta(z_{0:T}|\mathcal{S})||p_{ref}(z_{0:T}|\mathcal{S})] \quad (14)$$

This objective can be optimized directly through the conditional path distribution $p_\theta(z_{0:T}|\mathcal{S})$ via a DPO-style loss:

$$L_{\text{DPO-BridgePath}}(\theta) = -\mathbb{E}_{(\mathcal{S}, \mathcal{Y}^w, \mathcal{Y}^l) \sim \mathcal{D}_{energy}} \left[\log \sigma \left(\beta \mathbb{E}_{z_{1:T}^w \sim p_\theta(z_{1:T}^w|\mathcal{Y}^w, \mathcal{S})} \left[\log \frac{p_\theta(z_{0:T}^w|\mathcal{S})}{p_{ref}(z_{0:T}^w|\mathcal{S})}\right.\right.\right.$$
$$\left.\left.\left. - \log \frac{p_\theta(z_{0:T}^l|\mathcal{S})}{p_{ref}(z_{0:T}^l|\mathcal{S})}\right]\right)\right]$$
$$(15)$$

where $z_T^w = \mathcal{Y}^w$ and $z_T^l = \mathcal{Y}^l$.

## C.3 ADDRESSING INTRACTABILITY OF PATH SAMPLING AND LIKELIHOODS

Optimizing the above equation is challenging because:

- Sampling the full path $z_{1:T} \sim p_\theta(z_{1:T}|\mathcal{Y}, \mathcal{S})$ (the reverse bridge process pinned at $\mathcal{Y}$) is inefficient and potentially intractable during training.

- The path likelihoods $p_\theta(z_{0:T}|\mathcal{S})$ are also intractable.

We can substitute the reverse decompositions $p_\theta(z_{0:T}|\mathcal{S}) = p(z_0|\mathcal{S}) \prod_{t=1}^{T} p_\theta(z_{t-1}|z_t, \mathcal{S})$ (assuming $z_0$ is fixed given $\mathcal{S}$, so $p(z_0|\mathcal{S})$ might be a delta function or a simple prior $\mathcal{X}$). The log-likelihood ratio for a path then becomes a sum of single-step log-likelihood ratios:

$$\log \frac{p_\theta(z_{0:T}|\mathcal{S})}{p_{ref}(z_{0:T}|\mathcal{S})} = \sum_{t=1}^{T} \log \frac{p_\theta(z_{t-1}|z_t, \mathcal{S})}{p_{ref}(z_{t-1}|z_t, \mathcal{S})} \quad (16)$$

By utilizing Jensen's inequality and assuming uniform sampling of timesteps $t \sim \mathcal{U}(0, T)$, we can get a bound:

$$
\begin{aligned}
L_{\text{DPO-BridgeStep}}(\theta) \leq - \mathbb{E}_{(\mathcal{S}, \mathcal{Y}^w, \mathcal{Y}^l), t, z_{t-1,t}^w, z_{t-1,t}^l} \Bigg[ & \log \sigma \left( \beta T \left( \log \frac{p_\theta(z_{t-1}^w | z_t^w, \mathcal{S})}{p_{ref}(z_{t-1}^w | z_t^w, \mathcal{S})} \right. \right. \\
& \left. \left. - \log \frac{p_\theta(z_{t-1}^l | z_t^l, \mathcal{S})}{p_{ref}(z_{t-1}^l | z_t^l, \mathcal{S})} \right) \right) \Bigg]
\end{aligned}
\tag{17}
$$

Here, $z_{t-1,t}$ are sampled from $p_\theta(z_{t-1,t} | \mathcal{Y}, \mathcal{S})$.

### C.4 APPROXIMATION USING THE FORWARD PROCESS AND MODEL OBJECTIVE

Sampling from the reverse joint $p_\theta(z_{t-1}, z_t | \mathcal{Y}, \mathcal{S})$ is still difficult. Diffusion-DPO approximates the reverse process $p_\theta(x_{1:T} | x_0)$ with the forward noising process $q(x_{1:T} | x_0)$. For Markov Bridges, we are interested in the model's ability to predict the target sequence $\mathcal{Y}$ (or its properties) from an intermediate state $z_t$. The pre-training objective for the Markov Bridge model $\phi_\theta$ in EnerBridge-DPO is given by minimizing a loss related to predicting the target sequence $\mathcal{Y}$ given $z_t, \mathcal{S}, t$:

$$
\mathcal{L}_{\text{pretrain}_t}(\theta) = \lambda_t \mathbb{E}_{p(z_t | \mathcal{X}, \mathcal{Y})}[-v_t \mathcal{Y}^T \log \phi_\theta(z_t, \mathcal{S}, t)]
\tag{18}
$$

This is essentially a negative log-likelihood. The key insight from DPO is that the log-likelihood ratio $\log \frac{p_\theta(z_{t-1}|z_t, \mathcal{S})}{p_{ref}(z_{t-1}|z_t, \mathcal{S})}$ can be related to the difference in the "energies" or "losses" assigned by the models $p_\theta$ and $p_{ref}$. For Bridge-DPO, we adapt this by considering the "error" the model $\phi_\theta$ makes in predicting the target sequence $\mathcal{Y}$ from $z_t$. The term $\|\mathcal{Y} - \phi_\theta(z_t, \mathcal{S}, t)\|_2^2$ represents such an error term (e.g., L2 loss if $\phi_\theta$ predicts sequence embeddings, or it can be a proxy for negative log-likelihood if $\phi_\theta$ predicts probabilities).

The intermediate states $z_t^w$ and $z_t^l$ are sampled from the forward bridge process, conditioned on the prior sequence $\mathcal{X}$ (derived from $\mathcal{S}$) and implicitly on the target sequences $\mathcal{Y}^w$ and $\mathcal{Y}^l$ respectively. The EnerBridge-DPO simplifies this to $z_t \sim q(z_t | \mathcal{X}, \mathcal{S})$ for the purpose of the DPO loss, which is a common simplification in DPO-like objectives where the "noised" versions are generated from the data points.

### C.5 FINAL BRIDGE-DPO LOSS FORMULATION

Combining these ideas leads to the final Bridge-DPO loss function as:

$$
\begin{aligned}
\mathcal{L}_{\text{Bridge-DPO}}(\theta) = - \mathbb{E}_{(\mathcal{Y}^w, \mathcal{Y}^l) \sim \mathcal{D}, t \sim \mathcal{U}(0, T), \boldsymbol{z}_t^w \sim q(\boldsymbol{z}_t^w | \mathcal{X}, \mathcal{S}), \boldsymbol{z}_t^l \sim q(\boldsymbol{z}_t^l | \mathcal{X}, \mathcal{S})} \\
\log \sigma (-\beta T \omega(\lambda_t) (\|\mathcal{Y}^w - \varphi_\theta(\boldsymbol{z}_t^w, t)\|_2^2 - \|\mathcal{Y}^w - \varphi_{\text{ref}}(\boldsymbol{z}_t^w, t)\|_2^2) \\
- (\|\mathcal{Y}^l - \varphi_\theta(\boldsymbol{z}_t^l, t)\|_2^2 - \|\mathcal{Y}^l - \varphi_{\text{ref}}(\boldsymbol{z}_t^l, t)\|_2^2) )
\end{aligned}
\tag{19}
$$

This loss encourages $\phi_\theta$ to have a relatively lower error for preferred sequences ($\mathcal{Y}^w$) and/or a relatively higher error for dispreferred sequences ($\mathcal{Y}^l$) compared to the reference model $\phi_{ref}$, across the bridge timesteps. This derivation parallels the Diffusion-DPO approach by using model prediction errors as a proxy for the terms in the DPO objective, adapted to the Markov Bridge framework.

## D VISUALIZATION FOR PROTEIN FOLDING

To better show how EnerBridge-DPO designs low-energy sequences, we picked four protein complexes from our test set to look at. Figure 3 compares the folded structures of sequences designed by EnerBridge-DPO and Bridge-IF with the reference crystal structures. We also used BA-Cycle to predict the energy of these sequences.

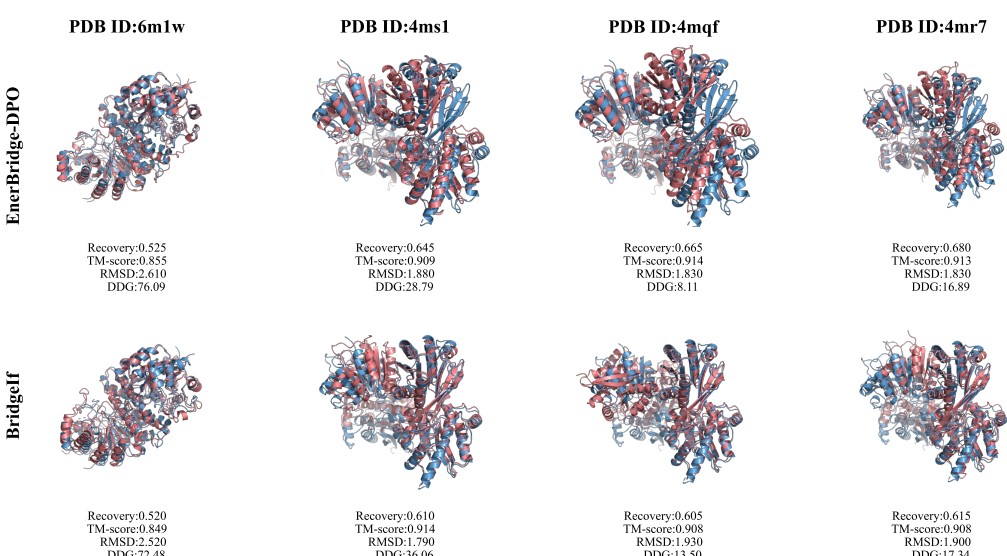

Figure 3: Folding comparison of designed sequences (in red) and the native sequences (in blue).

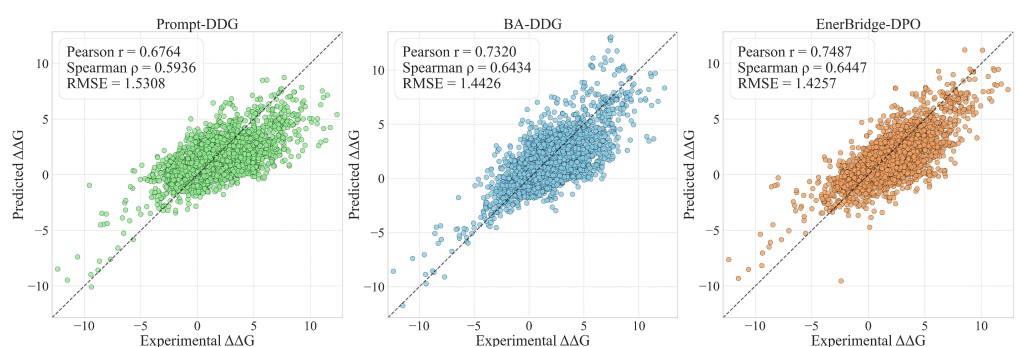

Figure 4: Comparison of correlations between experimental $\Delta\Delta G$ and predicted $\Delta\Delta G$.

## E    IMPLEMENTATION DETAILS

### E.1    STRUCTURE ENCODER

In our framework, the structure-conditioned prior sequence $\mathcal{X}$ is generated by a structure encoder $\mathcal{E}(\mathcal{S})$. In EnerBridge-DPO, we use PiFold as the structure encoder.

### E.2    PREFERENCE DATA GENERATION DETAILS

To construct the preference dataset $\mathcal{D}_{energy}$ for DPO fine-tuning, sequence pairs were carefully selected and constructed from the BindingGym and SKEMPI datasets. Specifically, for the BindingGym dataset, for each protein we selected the top $10\%$ of mutants with the highest scores as potential 'winners' and the bottom $10\%$ with the lowest scores as potential 'losers'. We then randomly paired sequences from these two groups, resulting in 47,297 million preference pairs. For the SKEMPI v2 dataset, we selected mutant sequences from the top $30\%$ and the bottom $30\%$ and randomly combined them to construct preference pairs, yielding approximately 3,744 preference pairs.

## F    VISUALIZATION FOR $\Delta\Delta G$ PREDICTION

Figure 4 presents a comparative analysis of the correlation between experimentally determined $\Delta\Delta G$ values and those predicted by different computational methods. The figure comprises three

scatter plots, corresponding to Prompt-DDG, BA-DDG, and our proposed EnerBridge-DPO model. In each scatter plot, the x-axis represents the experimental $\Delta\Delta G$ values, while the y-axis denotes the $\Delta\Delta G$ values predicted by the respective model. Ideally, the data points should cluster closely around the diagonal line, indicating a strong agreement between predicted and experimental values.

Visually, the plot for EnerBridge-DPO shows the strongest correlation with experimental data, with its data points appearing more tightly clustered around the diagonal compared to BA-DDG and Prompt-DDG. EnerBridge-DPO achieves higher Pearson and Spearman correlation coefficients and the lowest Root Mean Square Error (RMSE), indicating a smaller overall deviation between its predictions and the experimental results. Collectively, these metrics suggest that EnerBridge-DPO demonstrates superior performance in accurately predicting changes in binding free energy upon mutation compared to the other methods shown.

## G BROADER IMPACTS

EnerBridge-DPO can significantly accelerate the design of energetically stable proteins, promising advances in developing novel therapeutics and biotechnological solutions, while also deepening our fundamental understanding of protein science. However, thorough experimental validation of all computationally designed proteins is crucial to ensure their real-world safety and efficacy. The responsible development and deployment of this AI-driven technology are paramount to harness its substantial benefits for societal good, particularly in health and sustainability.

