# OpenReview forum: "EnerBridge-DPO: Energy-Guided Protein Inverse Folding with Markov Bridges and Direct Preference Optimization"
_ICLR.cc/2026/Conference — ICLR 2026 Conference Withdrawn Submission_

### Official Review · Reviewer_S1Fo · 2025-10-31

**Soundness:** 3
**Presentation:** 3
**Contribution:** 2
**Rating:** 4
**Confidence:** 3

**Summary:**

EnerBridge-DPO introduces an energy-guided protein inverse folding framework that integrates Markov Bridges with Direct Preference Optimization to generate low-energy, high-stability protein sequences while preserving structural fidelity.

**Strengths:**

+ Combines Markov Bridge generative modeling with Direct Preference Optimization, introducing energy-based fine-tuning into protein inverse folding for the first time.
+ Incorporates explicit energy constraints and ΔΔG prediction, aligning learned representations with biophysical energy landscapes.
+ Demonstrates lower energy, stable protein designs, and competitive recovery rates across multiple benchmarks with solid ablation analyses.

**Weaknesses:**

+ The model’s energy improvements rely on computational predictors (FoldX, Rosetta, BA-Cycle) without experimental or molecular dynamics confirmation.

+ DPO fine-tuning depends on precomputed or predicted energy scores, which may introduce bias and limit generalization to unseen proteins.
+ The paper lacks discussion on computational cost, hyperparameter sensitivity (e.g., β in DPO), and robustness across large or diverse protein complexes.

**Questions:**

1. How sensitive is EnerBridge-DPO to the β hyperparameter in the DPO term? Does higher β harm diversity?

2. Could the model generalize to de novo backbones not present in the training set?

3. Is the energy predictor differentiable and updated during DPO, or fixed as an external oracle?

---

### Official Review · Reviewer_DcbP · 2025-11-01

**Soundness:** 3
**Presentation:** 3
**Contribution:** 3
**Rating:** 4
**Confidence:** 4

**Summary:**

The paper introduces EnerBridge-DPO, a protein inverse-folding framework that unifies Markov bridge generative modeling with Direct Preference Optimization (DPO) and Boltzmann-aligned energy constraints.
The method attempts to address a current inverse-folding limitation — they optimize for sequence recovery but neglect thermodynamic stability. In this paper they focus on binding free energy of protein complexes.

EnerBridge-DPO proceeds in two stages:
Markov Bridge Pre-training: Trains AdaLN-Bias and Cross-Attention Adapter layers on top of frozen ESM2 weights to iteratively transition the predicted PiFold sequence into the true native sequence.

Bridge-DPO Fine-tuning: Fine-tunes the Markov Bridge model using DPO guided by ∆∆Gbind values for protein complexes. Here, DPO steers the bridge prediction towards a lower-energy sequence.

Experiments on BindingGym, SKEMPI, and PDB benchmarks show improved sequence recovery, lower predicted binding energies, and accurate ΔΔG prediction compared to existing baselines like ProteinMPNN and Bridge-IF.

**Strengths:**

Combination of Markov Bridge and DPO.
The combination of Markov bridges (for structured stochastic refinement) with DPO (for preference-based alignment) is new and elegant. The paper demonstrates a theoretically consistent formulation where probabilistic modeling (via bridge processes) and preference learning (via DPO) jointly improve protein design.

Interesting empirical results.
Ablation studies confirm the necessity of both DPO fine-tuning and energy supervision in various downstream results.

Clean mathematical formulation.
The paper provides a clear derivation of the bridge process and adapts the DPO loss to the inverse-folding domain, including justification of each term.

Computational efficiency.
Training with only T = 25 timesteps, a cosine noise schedule, and simple Adam/Noam optimization shows practical efficiency — feasible for wider adoption.

**Weaknesses:**

Energy-aware learning objective.
The inclusion of a Boltzmann-aligned energy loss attempts to ground the generative process in physical thermodynamics in order to make the model interpretable and biologically relevant. However, the assumption that model probability strongly correlates with free energy is a poor decision by the author for several reasons. The biggest reason is that the datasets used for ∆∆Gbind are very noisy datasets with heterogenous analytical methods used for data collection, they often use proxies rather than actually measuring ∆∆G, these measurements are often dependent on the temperature, buffer, etc making it hard to aggregate data between different labs and proteins.

Ablations on DPO temperature and preference data.
It is unclear how sensitive performance is to the β parameter or the construction of winner–loser pairs. More analysis here would help assess robustness.

Limited experimental scope.
Evaluation focuses mainly on sequence recovery and ΔΔGbind prediction; it would strengthen the paper to test downstream structure quality (e.g., AlphaFold2-refolded RMSD) or binding specificity. Using stability ∆∆G data would better demonstrate effectiveness of the DPO fine-tuning.

**Questions:**

Ambiguous difference in sequence pairs.
How many mutations are there between the positive and negative pairs in the BindingGym data used for DPO? I didn't see any stats for this in the paper. I know most of SKEMPI is single point mutations. Being able to show performance improvements for larger sequence differences (double/triple/etc mutants) compared to single point mutants might be an application that this method makes a meaningful improvement. Separating improvements for single point vs higher order mutants would improve benchmarking and evaluation of the method.

Better downstream evaluation results are needed.
While the method in itself (DPO + Markov bridges) is interesting, the downstream results are not very impressive, thus, questioning its utility. Table 2 shows modest improvements but the std is so large that I doubt it is statistically significant. Additionally, 3-fold cross validation is known to have serious data leakage and the performance improvements are marginal. In summary, seems like a lot of method development work for unimpressive results that are marginally better than baselines and are most likely the result of significant hyperparameter tuning. Please provide additional experiments that demonstrate true, unquestionable performance improvement in a downstream protein task.

Generalization:
Does this method work if you use ProteinMPNN or some other inverse folding framework? I don't see any ablations for changing the input prior sequence distribution.

---

### Official Review · Reviewer_aLtH · 2025-11-10

**Soundness:** 2
**Presentation:** 2
**Contribution:** 1
**Rating:** 2
**Confidence:** 4

**Summary:**

The authors propose EnerBridge- DPO, an inverse folding framework focused on generating low-energy, high-stability protein sequences. The method first pretrains a generative diffusion bridge mode that refines a structure-conditioned prior sequence from PiFold, then fine-tunes with an energy-guided DPO that prefers lower-energy sequences. An explicit energy constraint loss is introduced, compelling the model to learn and predict quantitative energy features.  The Experimental results demonstrate that EnerBridge-DPO designs protein complex sequences with lower energy compared to existing methods, while maintaining comparable sequence recovery.

**Strengths:**

1. The motivation is well grounded: effective sequence design should explicitly favor lower-energy sequences.

**Weaknesses:**

1. The methodological novelty appears very limited. The bridge-based generative component and several architectural/training choices (e.g., Markov Bridge formulation, PLM backbone with AdaLN-Bias and structural adapters, frozen base weights) closely track Bridge-IF [1], and the added DPO fine-tuning for lower energy reads as a relatively incremental extension rather than a fundamentally new framework.

2. The evaluation omits designability metrics, which are critical alongside stability/energy. Assessing only recovery/perplexity and energy leaves an incomplete picture of practical design performance. The authors should report standard designability measures (e.g., diversity, success rate under structure prediction, foldability metrics such as pLDDT/TM-score distributions for generated sequences) to substantiate claims about usable sequence design.

3. The ΔΔG prediction gains over BA-DDG in Table 3 are modest and appear incremental.

**Questions:**

For ΔΔG prediction, what fraction of the SKEMPI pairs overlap structurally or sequentially with training data used for pretraining or DPO?

**Details Of Ethics Concerns:**

A core component of the submission—the Markov Bridge architecture and its training setup—appears substantially derived from Bridge-IF [1], but the manuscript does not consistently or explicitly attribute these design choices and adopts a tone suggesting independent development. Three specific issues illustrate this concern:

1. Figure 2 is highly similar to Figure 2 in Bridge-IF, with the primary change being a rotation of the layout (left-to-right here versus top-to-bottom in [1]).

2. The methodological exposition in Section 3.2.3 closely parallels Bridge-IF’s Sections 4.3, 4.3.1, and 4.3.2: both employ a pretrained PLM as the backbone for approximating the reverse bridge process and tailor the Transformer blocks with  AdaLN-Bias for timestep conditioning alongside structural cross-attention adapters.

3. Both works freeze the base PLM weights during training.

Reference:

[1] Zhu et al., Bridge-IF: Learning Inverse Protein Folding with Markov Bridges. NeurIPS 2024.

---

> ### Author Response · Authors · 2025-11-13
> **A Rebuttal to Your Allegation of Plagiarism**
>
> Dear Reviewer,
>
> We take your concerns regarding academic integrity with the utmost seriousness, as this is a very serious allegation. We respectfully disagree with the reviewer's characterization that our work plagiarized Bridge-IF.
> We must first state that our contribution lies in the integration of DPO and explicit energy-based fine-tuning into the Bridge Diffusion framework. We have never claimed in the paper to have proposed Bridge Diffusion. On the contrary, we explicitly state in our introduction that our method is an improvement built upon Bridge-IF.
> It is possible that the description in certain sections was not sufficiently explicit in attributing the base architecture to Bridge-IF, which led to this misunderstanding. However, we respectfully disagree that this constitutes dishonesty, as we explicitly cited Bridge-IF as the foundation for our work in multiple key locations (e.g., the Introduction, Related Work, and the start of the Methodology section). Below, we will respond to your specific concerns regarding this issue and outline the revisions we will make to prevent any such misinterpretation.
> 1. Regarding the similarity of Figure 2 and Section 3.2.3
> You noted that our Figure 2 and the description in Section 3.2.3 are similar to those in Bridge-IF. This section was written to provide the necessary technical background and visual reference for the base model, so that our subsequently derived DPO objective (Eq. 7) and energy constraint (Eq. 9) would be understandable in context. We titled this subsection "Pre-trained Markov Bridge Model" and explicitly cited Bridge-IF within the text, stating: “We utilize the simplified reparameterized objective function derived in Bridge-IF for more effective training”. Our intent was to ensure the paper is self-contained; without this description, readers might find it difficult to understand how our novel contributions integrate with the Bridge Diffusion framework.
> We regret that our current presentation caused this confusion. In the revised manuscript, we will rename this subsection to "Preliminary: The Bridge-IF Base Model". We will also add explicit attribution phrases throughout the text and in the figure caption, such as: "Following the design in Bridge-IF, we adopt...".
> 2. Regarding freezing PLM weights
> We agree this is a shared design choice. This decision follows standard Parameter-Efficient Fine-Tuning practice, which was used in both works. We adopted this setting to ensure a fair comparison and to demonstrate that our performance gains originate from our DPO fine-tuning strategy, not from altering the base training paradigm.
>
> In summary, while we must respectfully disagree with the characterization of Figure 2 and Section 3 as plagiarism, we will thoroughly revise our manuscript's organization to strictly attribute the base architecture to Bridge-IF. This will serve to accurately highlight our true contribution—the novel DPO-based alignment framework—and eliminate any possibility of misunderstanding for the reader.

---

> ### Comment · Reviewer_aLtH · 2025-11-13
> **I maintain my plagiarism allegation regarding  the initial submission and defer to the ACs for a final decision**
>
> Dear authors,
>
> I appreciate that the revision will explicitly acknowledges that Figure 2 and the base model in Section 3.2.3 are derived from Bridge-IF.
>
> However, I do not accept the rebuttal’s explanation and I maintain my plagiarism concern regarding the initial submission. As a reviewer, it is my responsibility to flag plagiarism and document the basis for that assessment. My reasons remain as stated in the “Details of Ethics Concerns” section of my first-round review.
>
> I defer the final judgment to the Area Chairs. I recommend that the ACs closely compare:
>
> - The paper’s Figure 2 and Section 3.2.3 in the initial submission
>
> with
>
> - Bridge-IF’s Figure 2 and Sections 4.3, 4.3.1, and 4.3.2
>
> to determine whether the similarities meet the threshold for plagiarism.

---

### Note · Authors · 2026-01-10

I have read and agree with the venue's withdrawal policy on behalf of myself and my co-authors.